# The Economic Impact of Climate Change on Wheat and Maize Yields in the North China Plain

**DOI:** 10.3390/ijerph19095707

**Published:** 2022-05-07

**Authors:** Chunxiao Song, Xiao Huang, Oxley Les, Hengyun Ma, Ruifeng Liu

**Affiliations:** 1College of Economics and Management, Henan Agricultural University, Zhengzhou 450046, China; scx2011aqr@163.com (C.S.); ruifeng076@163.com (R.L.); 2College of Mechanical & Electrical Engineering, Henan Agricultural University, Zhengzhou 450046, China; huangxiao201998@163.com; 3Department of Economics and Finance, University of Waikato, Hamilton 3240, New Zealand; loxley@waikato.ac.nz

**Keywords:** climate change, extreme weather event, multi-level model, grain crop yield, village collective economy

## Abstract

Climate change has significantly affected agricultural production. As one of China’s most important agricultural production regions, the North China Plain (NCP) is subject to climate change. This paper examines the influence of climate change on the wheat and maize yields at household and village levels, using the multilevel model based on a large panel survey dataset in the NCP. The results show that: (i) Extreme weather events (drought and flood) would significantly reduce the wheat and maize yields. So, the governments should establish and improve the emergency service system of disaster warning and encourage farmers to mitigate the adverse effects of disasters. (ii) Over the past three decades, the NCP has experienced climate change that affects its grain production. Therefore, it is imperative to build the farmers’ adaptive capacity to climate change. (iii) Spatial variations in crop yield are significantly influenced by the household characteristics and the heterogeneity of village economic conditions. Therefore, in addition to promoting household production, it is necessary to strengthen and promote China’s development of the rural collective economy, especially the construction of rural irrigation and drainage infrastructures.

## 1. Introduction

Climatic conditions have always been an important factor shaping agricultural production. Climate change, especially in terms of extreme weather events, has exacerbated the fluctuations in food production and threatened world food security. In most part of China, increase in temperature is the main climate change issue reducing the major crop (wheat, rice, and maize) yields [1,2,3,4]. Increased extreme weather events associated with climate warms have exacerbated the decrease in food production in China. Since the 21st century (2000–2019), the average annual crop area affected by drought and flood were 17,966.6 and 10,011.1 thousand hectares, accounting for 11.3% and 5.3% of the total area, respectively. Crop yield loss due to drought has reached 26.39 million tons, and the crop loss rate has reached 4.7% [5,6]. As one of the most important agricultural production regions in China, the North China Plain (NCP) is subject to climate change and is often hit by extreme weather events, particularly drought [7].

Researchers have used econometric approaches to analyze the impact of climate change on grain production. Mendelsohn et al. [8] first proposed the Ricardian approach to analyze the climate change effect on farmland value (profit or net productivity of land). Liu et al. [9], Wang et al. [10], and Chen et al. [11] also employed the Ricardian approach to study the impact of climate change on China’s grain profit. However, the empirical results may be biased due to a few limitations, including the omission of irrigation variables in the model [10,12,13], the assumption of the unchanged price of grain and production inputs [14,15,16], and cost-free adaptation and adjustment implied in the models [17].

To address the limitations of the Ricardian approach, the production function approach was used to reveal an empirical relationship between climate factors and grain output in agricultural production, particularly in China [1,2,3,18,19,20,21,22]. However, most of the existing literature only focuses on long-term climate change such as changes in temperature and precipitation, while the studies on the impact of extreme weather events are scarce [23,24]. Furthermore, most previous literature uses macro data at the provincial or county level, which cannot effectively reflect the farmer’s behavioral selection characteristics or the village socio-economic characteristics, and their impact on crop yield. A village composed of farm households is the smallest administrative unit in rural China. The development of village collective economy plays a key role in ensuring food security, and it is an important guarantee for accelerating the building of a moderately prosperous society in all respects in rural areas [25,26]. Therefore, it is essential to capture the impact of climate change on grain yield at both the household and village levels [19,27].

To achieve the above goal, it is necessary to use a multilevel model to analyze large-scale survey data of farm households which typically adopt stratified multistage clustered sampling designs (with household level and village level). The multilevel analysis can model the clusters occurring at different levels of the sampling with nested random effects [28]. This study has shown that there were large spatial and temporal variations in climatic change factors in different growth stages of wheat and maize, and climatic factors in different growth stages have different effects on wheat and maize yield [3,29]. Meanwhile, this paper has also found that village heterogeneity plays a significant role in variation of gain yield, which likely indicates that developing rural village collective economy can reduce the negative effect of climate change on grain crop production in the NCP.

The rest of this paper is organized as follows: Section 2 briefly introduces the theoretical framework of multilevel model. Section 3 describes the sampling procedure and variables. The estimation results are presented in Section 4, and the final section concludes with some policy suggestions.

## 2. Theoretical Framework of Multilevel Model

The stratified sampling data with clustered characteristics show significant differences between different levels of data and high similarity among data at the same level. In this case, a regular ordinary least squares (OLS) model may result in misspecification by ignoring the average variation between groups. Therefore, a multi-level model (MLM) should be developed to deal with the heteroscedasticity caused by inter-dependent error terms and to estimate group-level averages by both fixed and random effects [30]. MLM decomposes the variance in the outcome into two components, one is attributed to the differences between individuals located in different groups and the other is related to the variation between individuals within the same group. This decomposition of variance into “between groups” and “within groups” corrects parameter estimation errors due to within-level sample similarity. Thus, this study uses the MLM to estimate the influencing factors of wheat and maize yields in the NCP at both household and village levels. The two types of MLM are introduced as follows.

### 2.1. Unconditional Means Model

The unconditional means model is an “empty model” that does not include any independent variables. It is reasonable to adopt MLM if individual respondents are clustered within groups and the variance of outcome in two levels are significantly different in a data structure. Assuming that $Y_{\mathrm{ij}}$ is grain yield measured for the ith farm plot of household in the jth village, the equations are as follows:(1)$$Y_{\mathrm{ij}}=\beta_{0j}+\varepsilon_{\mathrm{ij}}$$
(2)$$\beta_{0j}=\gamma_{00}+\mu_{0j}$$
where $\beta_{0j}$ represents the intercept term (the mean value of $Y_{\mathrm{ij}}$) for village j, and $\varepsilon_{\mathrm{ij}}$ is the residual for farm household in village j (an individual-level random component) in Equation (1). Equation (2) can be obtained by decomposing $\beta_{0j}$ into a fixed ($\gamma_{00}$) and a village-level random component ($\mu_{0j}$). Then, substituting (2) into (1) obtains Equation (3):(3)$$Y_{\mathrm{ij}}=\gamma_{00}+\mu_{0j}+\varepsilon_{\mathrm{ij}}$$
where $\gamma_{00}$ is the overall intercept or grand mean, $\mu_{0j}$ is a village-level random residual component indicating the average deviation from the grand mean for those farm households located in village j, and $\varepsilon_{\mathrm{ij}}$ remains the farm household-level residual. The usual assumption is that $\mu_{0j}\sim N\left(0,\sigma_{\mu }{}^{2}\right)$, $\varepsilon_{\mathrm{ij}}\sim N\left(0,\sigma_{\varepsilon }{}^{2}\right)$ and the $\mu_{0j}$ are independent from the $\varepsilon_{\mathrm{ij}}$. Thus, $\sigma_{\mu }{}^{2}$ and $\sigma_{\varepsilon }{}^{2}$ represent the between-group variance and the within-group variance, respectively. The intra-class correlation coefficient, ρ = $\frac{\sigma_{\mu }{}^{2}}{\sigma_{\varepsilon }{}^{2}+\sigma_{\mu }{}^{2}}$, is an indicator of the relative importance of village attributes, with larger values indicating a greater impact of village level on grain yield [31].

### 2.2. Random Intercept Model

The characteristics of farm households and villages that affect the grain yield remain unmeasured in Equation (3). Therefore, the variables of such characteristics are introduced to determine whether the between and within components of variation can be explained at the household and village levels. The random intercept model can be expressed as:(4)$$Y_{\mathrm{ij}}=(\gamma_{00}+\sum_{1}^{q}\beta_{0\mathrm{qj}}V_{0\mathrm{qj}}+\sum_{1}^{p}\beta_{\mathrm{pij}}X_{\mathrm{pij}})+(\mu_{0j}+\varepsilon_{\mathrm{ij}})$$
where, $X_{\mathrm{pij}}$ represents the independent variable of the farm household level, $V_{0\mathrm{qj}}$ represents the independent variable of the village level. Equation (4) consists of two parts: $\gamma_{00}+\sum_{1}^{q}\beta_{0\mathrm{qj}}V_{0\mathrm{qj}}+\sum_{1}^{p}\beta_{\mathrm{pij}}X_{\mathrm{pij}}$ as the fixed effects and $\mu_{0j}+\varepsilon_{\mathrm{ij}}$ as the random effects, and it can be expressed as:(5)Y=Xβ+ZU+e

Equation (5) is the general model of Equation (4). Where, Y is the observation variable; X is the design matrix of constant parameter β; Z is the design matrix of random effect U; and e is the random error. cov(Y)=V(θ). The logarithmic likelihood function of Equation (5) is given as:(6)$$\mathrm{lnL}\left(\beta ,\theta |Y\right)=-\ln\left|V\left(\theta \right)\right|-{\left(Y-X\beta \right)}^{\prime }V^{-1}\left(\theta \right)\left(Y-X\beta \right)$$

Maximum likelihood estimation of parameters can be obtained by maximizing Equation (6). That is, $\hat{\beta }\left(\theta \right)={\left(X^{\prime }V^{-1}\left(\theta \right)X\right)}^{-1}X^{\prime }V^{-1}\left(\theta \right)Y$ by fixing parameter θ. Then, plugging $\hat{\beta }\left(\theta \right)$ into L(β,θ|Y) can obtain the maximum likelihood estimation of θ.

## 3. Data Source and Empirical Model

### 3.1. Data Source

The NCP is one of China’s major grain production areas, accounting for approximately 75% and 35% of China’s wheat and maize outputs, respectively [6]. This region only grows winter wheat and summer maize. In recent years, this region has experienced evident climate change such as rising temperature and decreasing precipitation. The frequency of extreme weather events increases as the seasonal variation of precipitation becomes apparent. Specifically, flood often occurs in summer that receives 60% of the annual precipitation. Drought is often a serious threat in spring, autumn, and winter, especially in the areas without irrigation facilities [29,32]. The data used in this study are from a large-scale field survey of five provinces (Henan, Hebei, Shandong, Anhui, and Jiangsu) in the NCP.

To collect the data, stratified multi-stage cluster sampling was implemented. First, three counties were randomly chosen in each province using the following criterion. (i) The counties had experienced at least one episode of either severe drought or flood between 2010 and 2012. China’s national standard for natural disasters [33] categorizes the severity of droughts or floods into four levels: most severe, severe, moderate, and mild. A disaster year is when the government declares a warning of the most severe or severe flood or drought. (ii) The counties experienced at least one normal year in the past three years (2010, 2011 or 2012). Grain production usually experiences various weather shocks during any growing season; the term ‘normal year’ does not refer to a year without any weather shocks, but rather a year with no more than moderate weather shocks. Second, from each of the chosen counties, three townships were randomly selected to represent ‘good’, ‘medium’, and ‘poor’ local irrigation and drainage infrastructure conditions, respectively. Third, three villages were randomly selected from each township, and 10 households were randomly selected from each village for face-to-face interviews. Finally, from each household, two plots with grain production were randomly selected. Meteorological data were provided by National Meteorological Information Center (NMIC) (Data source: The China Meteorological Data Service Center (http://cdc.cma.gov.cn accessed on 15 May 2021), including the daily maximum temperature, minimum temperature, average temperature, and 24 h average precipitation recorded by the meteorological observatory in sample or adjacent counties.

As a result, the samples of winter wheat included 2261 plots of 1216 households, which were distributed in 123 villages (or 41 townships, 14 counties) of five provinces (Table 1). The samples of summer maize covered 1769 plots of 1028 households, distributed in 117 villages (or 40 townships, 14 counties) in five provinces (Table 2). Among the 14 case study counties, 10 suffered from drought disaster, and 4 suffered from flood disaster. The regional (provincial and county) distribution of all samples is shown in Figure 1.

The 12 main growth stages of winter wheat are seedling emergence, three-leaf, tillering, overwintering, reviving, jointing, booting, heading, anthesis, grain-filling, wax ripeness and mature. This study separated the overwintering stage from the vegetative stage (Firstly, this stage is a special stage of winter wheat to stop growing, which is quite important to store energy; secondly, China boasts the distinctive differences in regions and climate, especially the winter temperature change is more remarkable [18]); it also divided the whole growth period of winter wheat into three major growth stages: the overwintering stage from seedling emergence to reviving (generally from mid-October to mid-February in the following year), the vegetative stage from reviving to heading (generally from the mid-February to mid-April), and the reproductive stage from anthesis to maturity (generally from mid-April to early June). Similarly, the 12 main growth stages of summer maize include the stages of seedling, three-leaf, jointing, flare opening, tasseling, flowering, silking, filling, milk ripening, wax ripening, and full ripening. They were divided into three major growth stages: the vegetative stage from sowing to jointing (generally from mid-June to mid-July, about 20–30 days), the concurrent stage from jointing to silking (generally from mid-July to mid-August, about 27–30 days), and the reproductive stage from silking to full ripening (generally from mid-August to late September, about 40–60 days) [34].

Table 3 shows the climatic trend of various growth stages of winter wheat and summer maize in the NCP. In general, the overall climate change in the sample area was increasing temperature and precipitation from 1981 to 2010. The rangeability was inconsistent with temperature and precipitation over different grain growth stages. The warming trend during the winter was the most prominent, which further proves that the warming trend is most significant in the winter among the four seasons [35]. Precipitation increased the most during the whole growth period of summer maize, indicating that precipitation increase was most significant in the summer among the four seasons.

### 3.2. Empirical Model and Variables

Production inputs and economic and social institutional factors should be incorporated into the model; meanwhile, the factors of long-term climate change and extreme weather events should be included in the production function model. The C-D-C production function equation, which is the extension of Equation (4), is specified as:(7)$$\begin{array}{ll} \ln(Y_{\mathrm{ij}})= & \beta_{0}+\beta_{1}C_{\mathrm{ij}}+\beta_{2}D_{\mathrm{ij}}+\beta_{3}{\mathrm{DL}}_{\mathrm{ij}}+\beta_{4}\ln(I_{\mathrm{ij}})+\beta_{5}L_{\mathrm{ij}}\\ & +\beta_{6}H_{\mathrm{ij}}+\beta_{7}V_{\mathrm{ij}}+\beta_{8}V_{\mathrm{ij}}D_{\mathrm{ij}}+\beta_{8}V_{\mathrm{ij}}{\mathrm{DL}}_{\mathrm{ij}}+T+\mu_{0j}+\varepsilon_{\mathrm{ij}} \end{array}$$

This study independently investigated the effects of climate change and the household and village attributes on the yields of winter wheat and summer maize by Equation (6), respectively. The dependent variable Y_ij_ refers to crop yield, which is measured as the wheat or maize output per hectare. As shown in Table 4, the average yields of wheat and maize in the farm plots were 6400 kg and 6615.1 kg per hectare, respectively.

The variable of long-term climate change C_ij_ examined in this paper includes average daily temperature and precipitation over the past three decades (1981–2010). For winter wheat, the daily average temperature was only 5.2 °C in the over-wintering stage, and it was 20.4 °C in the reproductive stage. For summer maize, the daily average temperature could be above 20 °C, and the precipitation was more than 100 mm at different growth stages (Table 4).

The second climate indicator is extreme weather event, including the variables of county-level disaster D_ij_ and farm plot disaster DL_ij_. There are two county-level disasters, which are D_D_ for the severe drought year and D_F_ for the severe flood year. In the past three years (2010–2012), 24.7% of counties that grew wheat suffered from drought, and 25.1% and only 8.2% of counties that grew maize suffered from drought and flood, respectively. There are four types of farm plot disasters, which are DL_D_ for farm plots suffering from drought, DL_F_ for farm plots suffering from flood, DL_R_ for farm plots suffering from continuous rain, and DL_w_ for farm plots suffering from strong wind. In the past three years, 40.5% of farm plots growing wheat suffered from drought, 7–8% suffered from continuous rain or strong wind, while 2.7% suffered from flood, indicating that drought was the most frequent disaster during wheat planting. During the same period, 36% of farm plots that grew maize suffered from drought, 16% suffered from strong wind, and 15.5% suffered from flood. This shows that drought was the most frequent disaster during maize planting, and the risk of strong wind and flood should not be underestimated.

Three variables represent farmland plot characteristics. (i) The farmland areas L_1_ were relatively small, with an average farm area of only 0.21 ha and 0.19 ha for wheat and maize (Table 4), respectively, which indicates the formation of tiny plots and scattered planting; (ii) Most of the farmland topography L_2_ is flat land, and only less than 3% and 5% of farmers chose to grow wheat and maize in the mountains, respectively; (iii) Compared with the overall land quality of village, the farmland quality L_3_ is divided into three categories, which are low-quality, middle-quality and high-quality land. The majority of plots (68%) were of medium quality, 10% were of low-quality, and 20% were of high-quality.

$I_{\mathrm{ij}}$ is a set of production input variables, covering the fertilizer cost I_1_, pesticide cost I_2_, machinery cost I_3_, labor input I_4_, and irrigation water I_5_ at the plot level. Table 4 shows that among these costs, the average input costs of fertilizer were the highest with 2863.3 yuan and 2442.8 yuan per hectare, respectively, for wheat and maize. The cost of machinery was the second highest with 1678.4 yuan and 1248.3 yuan per hectare for wheat and maize, respectively, while the labor input costs of wheat and maize were 36.3 and 60.9 adult days per hectare. Thus, there might be a substitution relationship with machinery and labor. The irrigation water reached 1760.9 m^3^ and 1730.1 m^3^ for wheat and maize, respectively, indicating that the grain grown in the NCP is mainly irrigated rather than rainfed.

Farm household’s characteristics (H_ij_) include variables as follows. (i) The assets possessed by the household (H_1_), which are measured as the value of the durable goods. The average value of durable goods of sample households was 9700 yuan; (ii) H_2_ represents the education level of the household head, where the average education was 6.9 years. (iii) H_3_ represents the production and technical training, and about 25% of household members received such training.

Three variables were used to measure village characteristics. (i) V_1_ refers to village collective enterprise. The average village collective enterprise was only about 0.1, which means many villages did not have such enterprises. (ii) V_2_ refers to the ratio of irrigation area to total cultivated area in the village, which is more than 80%. (iii) V_3_ is the distance between the village committee and the nearest road above the township level with an average distance of 1.4 km.

In addition, this study used year dummies, T_2011_ (1 = 2011, otherwise = 0) and T_2012_ (1 = 2012, otherwise = 0), to control technological advances or other unobservable factors that change over time.

## 4. Results and Analyses

### 4.1. The Unconditional Means Model

Table 5 shows the estimated results of the unconditional means model with maximum likelihood estimation. The intra-class correlation coefficient ρ is 0.384, indicating 38.4% and 61.6% of the variation in wheat yield were caused by the village attributes and household attributes, respectively. Similarly, the inter-class correlation coefficient ρ is 0.238, indicating that 23.8% and 76.2% of the variation in maize yield were caused by the village attributes and household attributes, respectively. Due to the different samples in wheat and maize, the variation was different between villages and households. Moreover, a fairly large part of variation in the wheat and maize yields in the NCP was on the village level. Therefore, it is helpful to adopt a multilevel model to improve the accuracy of parameter estimation results.

### 4.2. The Random Intercept Model

According to Equation (6), the estimated results of the influence of climate factors and other factors on wheat and maize yields are shown in Table 6 and Table 7. Model I only included climate change variables and year dummy variables. Then, farmland attributes and production input variables were incorporated into Model II. Finally, village attributes were incorporated into Model III.

#### 4.2.1. The Determinants of Winter Wheat Yield

Table 6 displays the results of the three model for winter wheat yield. The likelihood function ratio, LR = 2[Ln(L_R2_) − Ln(L_R1_)] = 143.5, is greater than critical $\chi_{0.01}^{2}(12)=26.22$, meaning that farmland attributes and production input variables had significant impact on the variation in winter wheat yield. Furthermore, the AIC value is −3760.326 in Model II, which is less than AIC of −3640.829 in Model I, meaning that the better the overall fitting of Model II according to the information criteria, the smaller the AIC value, and the better the overall fitting of the model.

Similarly, Model III shows a better overall fitting than Model II. The likelihood function ratio LR = 2[Ln(L_R3_) − Ln(L_R2_)] = 54.17 is greater than critical $\chi_{0.01}^{2}(9)=21.67$, suggesting that village attributes had significant impact on the variation in winter wheat yield. Meanwhile, AIC of −3798.5 in Model III is smaller than AIC of −3760.326 in Model II. According to Table 6, the following conclusions can be drawn:

Firstly, the heterogeneity of wheat yield was not only caused by household characteristics, but also determined by differences in village economies, except for farmland characteristics and production input factors. For example, the variance of village level is $\sigma_{\mu }{}^{2}$ = 0.103 (Table 6, row 36) with introducing explanatory variables (Model III), which is less than the variance of $\sigma_{\mu }{}^{2}$ = 0.118 (Table 5, row 1) in the unconditional means model without the introduction of explanatory variables. It indicates that social and economic factors at the village level, such as collective economy (number of collective enterprises), irrigation condition (ratio of irrigation area), and traffic condition (distance between the village committee and the nearest road above the township level), could explain 12.7% ($\frac{\sigma_{\mu }{}^{2}\left(\mathrm{unconditional}\;\mathrm{means}\;\mathrm{model}\right)-\sigma_{\mu }{}^{2}\left(\mathrm{random}\;\mathrm{intercept}\;\mathrm{model}\right)}{\sigma_{\mu }{}^{2}(\mathrm{unconditional}\;\mathrm{means}\;\mathrm{model})}=\frac{0.118-0.103}{0.118}=0.127$) of the variation in wheat yield at the village level. Household attributes could explain 6.9% ($\frac{\sigma_{\varepsilon }{}^{2}\left(\mathrm{unconditional}\;\mathrm{means}\;\mathrm{model}\right)-\sigma_{\varepsilon }{}^{2}\left(\mathrm{random}\;\mathrm{intercept}\;\mathrm{model}\right)}{\sigma_{\varepsilon }{}^{2}(\mathrm{unconditional}\;\mathrm{means}\;\mathrm{model})}=\frac{0.189-\;0.176}{0.189}=0.069$) of variation in wheat yield, which was apparently and substantially smaller than village attributes.

Secondly, the effect of long-term climate change on wheat yield varied across different wheat growth stages. The increase of average temperature significantly promoted wheat production during the overwintering stage. For example, wheat yield would significantly increase by about 8% if temperature increased by 1 °C (row 1, Table 6). However, the increase of average temperature resulted in an obvious decrease in wheat yield during the vegetative stage. Specifically, wheat yield would significantly decline by 6.2–8.6% if average temperature improved by 1 °C (row 3, Table 6). These results indicate that the proper increase in winter temperature has a positive effect on winter wheat yield, while the increase in spring temperature can lead to a decrease of winter wheat yield, which is consistent with some previous studies [36]. This is probably because that the shortened growth period and warming temperature contribute to the increase of productive tiller [37,38].

Thirdly, the occurrence of extreme weather events had significantly negative impact on wheat yield. County-level drought significantly reduced wheat yield. Farm-level disasters also had led to obvious reduction in wheat yield. In particular, drought, flood, continuous rain, and strong wind at the farm level would reduce wheat yield by about 10%, 6%, 16%, and 9%, respectively, holding constant of other variables (rows 8–11, Table 6). Farm-level disasters showed a greater negative impact on wheat yield than county-level disasters.

Fourthly, production input significantly affected the wheat yield. The elasticity of labor is −0.016 with other input factors unchanged (row 21, Table 6). The sign of labor elasticity is negative, which is consistent with previous empirical studies [39]. The possible reasons are two-fold: on the one hand, there is a significant substitution relationship between labor input and machinery input, which leads to multiple collinearities contributing to the unreasonable estimated economic value of labor output elasticity; on the other hand, there is too much surplus labor force in agricultural production in China. The scattered and limited arable farmland will further increase surplus rural labor force, while labor has not fully flowed in the market. Therefore, it is more valuable of focusing on the quality of labor than the quantity of labor to improve grain yield and farmers’ income. Moreover, irrigation water could significantly and slightly promote wheat yield (0.004, row 22, Table 6), showing that wheat yield only increased by 0.004% with increasing irrigation water by 1%. The result means that the input of irrigation water is lack of elasticity.

Fifthly, the social and economic characteristics of households had a significant impact on wheat yield. As expected, education and participation in production and technology training programs played an important role in promoting wheat yield. In particular, the wheat yield would significantly increase by 0.2% if the schooling year of the household head increased by one (row 24, Table 6). Similarly, the wheat yield would significantly increase by 1.2% if farmers had previously undertaken a production and technique training (row 25, Table 6).

Finally, the social and economic characteristics of village could mitigate the loss of wheat yield under extreme weather events, especially drought disaster. For example, the wheat yield decreased by 19.7% if farm plot occurred drought disaster, but it only decreased by 1% if the distance shortened by 1 km between the village committee and the nearest township-level road. Moreover, the wheat yield decreased by 8.4% and 19.7%, respectively, when county-level drought and farm plot drought occurred simultaneously. However, in the case of drought, the wheat yield could significantly improve by 0.1% if irrigation proportion rise by 1%.

#### 4.2.2. The Determinants of Summer Maize Yield

Table 7 displays the determinant regression results of summer maize yield using three models. Similarly, according to the likelihood function ratio LR and information criterion AIC, Model III is more preferable than other models. The empirical results of Table 7 are similar to those in Table 6, but there are several differences as follows.

Firstly, the heterogeneity of maize yield was shaped by village attributes. In particular, the village level variance of $\sigma_{\mu }{}^{2}$ = 0.133 (row 36, Table 7) with introducing explanatory variables (Model III) is less than the variance of $\sigma_{\mu }{}^{2}$ = 0.173 (row 1, Table 5) without introducing explanatory variables (unconditional means model). This indicates that the social and economic characteristics of village, such as the village collective enterprise, the proportion of village irrigation area, and the distance between the village committee and the nearest road above the township level, could explain 23.1% ($\frac{\sigma_{\mu }{}^{2}\left(\mathrm{unconditional}\;\mathrm{means}\;\mathrm{model}\right)-\sigma_{\mu }{}^{2}\left(\mathrm{random}\;\mathrm{intercept}\;\mathrm{model}\right)}{\sigma_{\mu }{}^{2}(\mathrm{unconditional}\;\mathrm{means}\;\mathrm{model})}=\frac{0.173-0.133}{0.173}=0.231$) of the variation in maize yield at the village level. Household attributes could explain 4.1% ($\frac{\sigma_{\varepsilon }{}^{2}\left(\mathrm{unconditional}\;\mathrm{means}\;\mathrm{model}\right)-\sigma_{\varepsilon }{}^{2}\left(\mathrm{random}\;\mathrm{intercept}\;\mathrm{model}\right)}{\sigma_{\varepsilon }{}^{2}(\mathrm{unconditional}\;\mathrm{means}\;\mathrm{model})}=\frac{0.555-0.532}{0.555}=0.041$) of variation in maize yield, which was apparently and considerately smaller than village attributes.

Secondly, the occurrence of extreme weather events had a significantly negative impact on maize yield. For example, the maize yield decreased by 12.7% and 14.2% under drought and flood disasters at the county level, respectively. The disasters occurring on farmland plot led to more loss in maize yield than those at the county level. Specifically, drought, flood, continuous rain, and strong wind reduced maize yield by 13.6%, 22.4%, 12.2%, and 9.8%, respectively.

Thirdly, production input significantly affected the maize yield. The sign of labor elasticity is negative, and the reasons have been mentioned above. Pesticide and irrigation water input could significantly promote maize yield. Specifically, the maize yield would increase by 0.032% and 0.018% for every 1% increase in pesticide input and irrigation water, respectively (rows 20 and 23, Table 7).

Finally, the social and economic characteristics of village could mitigate the loss of maize yield under extreme weather events. The coefficients of the cross term between village collective enterprise and drought disaster reveal that the negative impact of drought disaster on maize yield would decrease dramatically if the village had more collective enterprises. For example, the maize yield declined by 48.9% if drought occurred on the farmland plot, but it only decreased by 11% for one additional collective enterprise. Furthermore, the maize yield could decrease by 9.1% under county-level drought, but it could increase by 12.8% for an additional collective enterprise.

## 5. Conclusions and Discussion

Based on the data of 6749 wheat plots and 5212 maize plots of farm households over 2010–2012, this paper adopted a multilevel model to analyze the impact of long-term climate change and extreme weather events on the wheat and maize yields in the NCP. It also considered village social and economic conditions, social and economic characteristics of the household, production inputs, and farmland plot characteristics as the influencing factors of wheat and maize yields. The findings of this study suggest the following conclusion and discussion.

### 5.1. Conclusions

There are three main findings and conclusions in this study.

Firstly, spatial variations in crop yield are significantly influenced by the heterogeneity of village economic conditions. The explained variation in crop yield is much higher at the village level than at the household level. The social and economic characteristics of the village have a positive effect on crop yield and mitigate the loss of crop yield under extreme weather events. Therefore, under China’s dual-level management system of integration of unification and separation in rural area, it is necessary to strengthen household production behavior and improve village collective economy.

Secondly, the arid and semi-arid region of NCP has been experiencing climate change, affecting grain production over the past three decades. The effects of long-term climate variables on winter wheat and summer maize yields vary across the growth stages. Therefore, it might be time to think of making agricultural production adapt to climate change. For example, it may need to adjust the planting system, change crop varieties, and build households’ adaptive capacity to climate change.

Thirdly, extreme weather events are more likely than long-term climate change to reduce the wheat and maize yields. The negative impact of extreme weather events on crop yield is more serious and immediate. These findings suggest that the governments should establish and improve the disaster service and coping system for grass-roots units. Specifically, it is essential to use modern information technology to improve the monitoring, forecasting, and warning of agricultural disasters and provide and publicize disaster early warning and response information timely. At the same time, agricultural technical guidance and financial support should be provided for disaster prevention and control, enabling farmers to minimize crop production loss. The households also need to prepare for extreme weather events.

### 5.2. Discussion

In dialogue with the existing literature, we respond to, and confirm, the scholarly view that the role of the village collective economy plays a key role in ensuring food security [25,26]. We have shown that the village collective economy can mitigate the loss of crop yield under extreme weather events. In addition, the impact of extreme weather events on crop yield should be emphasized when revealing an empirical relationship between climate factors and grain output in agricultural production [23,24]. Since the negative impact of extreme weather events on crop yield is more serious and immediate than long-term climate change.

As usual, this paper still has some limitation. Due to data unavailable, for example, the impact of the township or even the county economic levels cannot be taken into consideration; long-term temperature and precipitation at county-level data have to be used for plot-level modelling. For the future study, it be better incorporating long-term plot-level temperature and precipitation into plot-level modelling by taking into consideration of county- and -township-level economic impacts.

## Figures and Tables

**Figure 1 ijerph-19-05707-f001:**
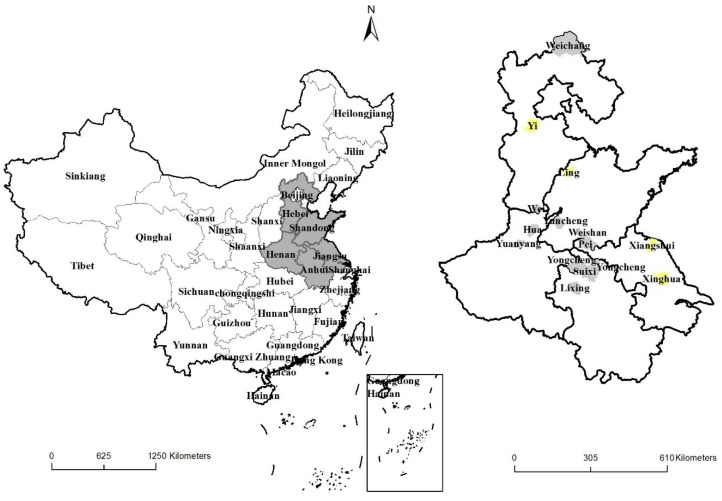
Location of five provinces in the NCP (**left**) and 14 sample counties (**right**).

**Table 1 ijerph-19-05707-t001:** The sample distribution of winter ***wheat*** for the NCP.

Province	County	No. of Households	No. of Plots	Disaster Type	Disaster/Normal Year
Henan	Yuanyang	90	167	D	2011/2012
	Huanxian	90	160	D	2011/2012
	Yongcheng	90	176	D	2011/2012
Hebei	Weixian	90	164	D	2011/2012
	Yixian	56	93	F	2012/2011
Shandong	Lingxian	90	167	F	2012/2011
	Yuncheng	90	174	D	2011/2012
	Huishan	90	159	D	2011/2012
Jiangsu	Xinghua	89	160	F	2011/2012
	Xiangshui	90	171	F	2012/2011
	Peixian	81	146	D	2011/2012
Anhui	Yongqiao	90	175	D	2011/2012
	Suixi	90	172	D	2011/2012
	Lixin	90	177	D	2011/2012
Total	14	1216	2261	-	-

Notes: D and F stand for drought and flood, respectively.

**Table 2 ijerph-19-05707-t002:** The sample distribution of summer ***maize*** for the NCP.

Province	County	No. of Households	No. of Plots	Disaster Type	Disaster/Normal Year
Henan	Yuanyang	72	128	D	2011/2012
	Huanxian	90	159	D	2011/2012
	Yongcheng	62	113	D	2011/2012
Hebei	Weixian	90	164	D	2011/2012
	Yixian	90	162	F	2012/2011
Shandong	Lingxian	90	167	F	2012/2011
	Yuncheng	90	172	D	2011/2012
	Huishan	90	159	D	2011/2012
Jiangsu	Xinghua	11	12	F	2011/2012
	Xiangshui	82	89	F	2012/2011
	Peixian	63	93	D	2011/2012
Anhui	Yongqiao	67	119	D	2011/2012
	Suixi	62	106	D	2011/2012
	Lixin	69	126	D	2011/2012
Total	14	1028	1769	-	-

Notes: D and F stand for drought and flood, respectively.

**Table 3 ijerph-19-05707-t003:** Climatic trend rate of major crop growth stages in the NCP (1981–2010).

Crop Growth Stages	Daily Average Temperature (°C/10a)	Average Precipitation (cm/10a)
Winter wheat:		
Overwintering stage	0.519	0.115
Vegetative stage	0.675	0.66
Reproductive stage	0.305	1.137
Summer maize:		
Vegetative stage	0.319	1.601
Concurrent stage	0.153	2.25
Reproductive stage	0.229	1.229

The sample data comes from meteorological observation stations in 14 wheat and maize producing counties. Regressed the meteorological variables and time variables of each sample county linearly, and weighted average of all regression coefficients to obtain the annual change rate, which multiply by 10 to obtain climatic trend rate.

**Table 4 ijerph-19-05707-t004:** Summary statistics of variables used.

Variables	Definition	Winter Wheat		Summer Maize	
		Mean	S.D.	Mean	S.D.
Explained variables:					
Grain yield (Y)	Kg/ha	6400	1176	6615	1535
Explanatory variables:					
The variables of long-run climate change (wheat):					
Daily avg temperature in overwintering stage (T_wheat1_)	°C	5.22	1.19	-	-
Total avg precipitation in overwintering stage (P_wheat1_)	cm	8.40	2.89	-	-
Daily avg temperature in vegetative stage (T_wheat2_)	°C	9.67	1.47	-	-
Total avg precipitation in vegetative stage (P_wheat2_)	cm	7.64	4.05	-	-
Daily avg temperature in reproductive stage (T_wheat3_)	°C	20.38	0.81	-	-
Total avg precipitation in reproductive stage (P_wheat3_)	cm	8.53	2.46	-	-
The variables of long-run climate change (maize):					
Daily avg temperature in vegetative stage (T_maize1_)	°C	-	-	26.13	0.49
Total avg precipitation in vegetative stage (P_maize1_)	cm	-	-	10.73	3.56
Daily avg temperature in concurrent stage (T_maize2_)	°C	-	-	27.12	0.41
Total avg precipitation in concurrent stage (P_maize2_)	cm	-	-	16.95	2.74
Daily avg temperature in reproductive stage (T_maize3_)	°C	-	-	23.33	1.20
Total avg precipitation in reproductive stage (P_maize3_)	cm	-	-	16.93	3.20
Extreme weather events:					
If it occurred drought disaster at the county-level (D_D_)	1 = Yes; 0 otherwise	0.25	0.43	0.25	0.43
If it occurred flood disaster at the county-level (D_F_)	1 = Yes; 0 otherwise	-	-	0.08	0.27
If it occurred drought disaster on farm plot (DL_D_)	1 = Yes; 0 otherwise	0.41	0.49	0.36	0.48
If it occurred flood disaster on the farm plot (DL_F_)	1 = Yes; 0 otherwise	0.03	0.16	0.16	0.36
If it occurred continuous rain disaster on farm plot (DL_R_)	1 = Yes; 0 otherwise	0.08	0.26	0.04	0.19
If it occurred strong wind disaster on farm plot (DL_w_)	1 = Yes; 0 otherwise	0.08	0.27	0.16	0.37
Farmland plot characteristics:					
Farmland area (L_1_)	Hectare	0.21	0.18	0.19	0.13
Farmland topography (L_2_)	1 = flat land; 0 = otherwise	0.98	0.14	0.06	0.24
Low quality of farmland (L_31_)	1 = Yes; 0 otherwise	0.11	0.31	0.12	0.33
Medium quality of farmland (L_32_)	1 = Yes; 0 otherwise	0.70	0.46	0.67	0.47
High quality of farmland (L_33_)	1 = Yes; 0 otherwise	0.19	0.39	0.21	0.41
Production inputs:					
Fertilizer cost (I_1_)	Yuan/ha	2863.29	1246.98	2442.79	1063.44
Pesticide cost (I_2_)	Yuan/ha	331.24	263.68	472.71	321.17
Machinery cost (I_3_)	Yuan/ha	1678.38	577.16	1248.26	800.56
Labor input (I_4_)	Adult days/ha	36.26	34.52	60.90	63.69
Irrigation water (I_5_)	m^3^/ha	1760.88	1753.53	1730.09	2279.84
Household’s characteristics:					
Asset of household (H_1_)	Durable goods (10^3^ yuan)	9.67	19.24	9.86	19.48
Education of household head (H_2_)	Attending year	6.91	3.19	6.93	3.11
Producing/technical training (H_3_)	If attending (1 = Yes; 0 otherwise)	0.27	0.45	0.24	0.42
Village’s characteristics					
Collective enterprise (V_1_)	Number of collective enterprises	0.08	0.55	0.13	0.768
Ratio of irrigation area to total cultivated area (V_2_)	%	83.85	23.71	83.17	27.88
Distance between the village committee and the nearest road above the township level (V_3_)	Km	1.36	1.55	1.38	1.58
Year dummy variables:					
2011 (T_2011_)	1 = Yes; 0 otherwise	0.33	0.47	0.33	0.47
2012 (T_2012_)	1 = Yes; 0 otherwise	0.33	0.47	0.33	0.47
Observations	-	6749		5212	

**Table 5 ijerph-19-05707-t005:** The estimated results of unconditional means model.

Variance Decomposition	Winter Wheat		Summer Maize	
	Coefficient	S.D.	Coefficient	S.D.
Variance of village level (between-group variance)	0.118	0.008	0.173	0.014
Variance of household level (within-group variance)	0.189	0.002	0.555	0.005
Intra-class correlation coefficient ρ	0.384	-	0.238	-

**Table 6 ijerph-19-05707-t006:** The estimated results of influencing factors of winter ***wheat*** yield.

Variables	Model I	Model II	Model III
T_wheat1_	0.080 ** (0.032)	0.079 ** (0.031)	0.088 *** (0.032)
P_wheat1_	−0.087 *** (0.025)	−0.088 *** (0.025)	−0.097 *** (0.026)
T_wheat2_	−0.068 * (0.037)	−0.062 * (0.036)	−0.086 ** (0.038)
P_wheat2_	0.054 *** (0.021)	0.052 ** (0.021)	0.065 *** (0.022)
T_wheat3_	0.041 (0.032)	0.036 (0.032)	0.051 (0.032)
P_wheat3_	−0.002 (0.015)	0.005 (0.015)	−0.002 (0.015)
D_D_	−0.032 *** (0.011)	−0.033 *** (0.011)	−0.084 *** (0.022)
DL_D_	−0.096 *** (0.006)	−0.094 *** (0.006)	−0.197 *** (0.02)
DL_F_	−0.057 *** (0.015)	−0.056 *** (0.014)	−0.055 *** (0.014)
DL_R_	−0.158 *** (0.01)	−0.161 *** (0.009)	−0.160 *** (0.009)
DL_W_	−0.088 *** (0.009)	−0.084 *** (0.009)	−0.086 *** (0.009)
T_2011_	0.036 *** (0.01)	0.035 *** (0.009)	0.034 *** (0.009)
T_2012_	−0.033 *** (0.005)	−0.033 *** (0.005)	−0.033 *** (0.005)
L_1_	−	0.005 (0.015)	0.003 (0.015)
L_2_	−	−0.009 (0.016)	−0.010 (0.016)
L_32_	−	0.06 *** (0.007)	0.06 *** (0.007)
L_33_	−	0.083 *** (0.009)	0.083 *** (0.009)
ln(I_1_)	−	0.007 (0.005)	0.006 (0.005)
ln(I_2_)	−	−0.002 (0.002)	−0.002 (0.002)
ln(I_3_)	−	−0.003 (0.006)	−0.003 (0.006)
ln(I_4_)	−	−0.016 *** (0.004)	−0.016 *** (0.004)
ln(I_5_)	−	0.004 *** (0.001)	0.004 *** (0.001)
H_1_	−	0.0001 (0.0001)	0.0001 (0.0001)
H_2_	−	0.002 ** (0.001)	0.002 ** (0.001)
H_3_	−	0.012 ** (0.006)	0.012 ** (0.006)
V_1_	−	−	0.011 (0.015)
V_2_	−	−	−0.0001 (0.0003)
V_3_	−	−	0.002 (0.007)
V_1_ × D_D_	−	−	−0.004 (0.009)
V_1_ × DL_D_	−	−	0.006 (0.009)
V_2_ × D_D_	−	−	0.001 *** (0.0002)
V_2_ × DL_D_	−	−	0.001 *** (0.0002)
V_3_ × D_D_	−	−	0.001 (0.004)
V_3_ × DL_D_	−	−	0.01 *** (0.004)
Cons.	8.542 *** (0.421)	8.501 *** (0.418)	8.447 *** (0.415)
$\mathrm{Variance}\;\sigma_{\mu }{}^{2}$	0.106 (0.007)	0.105 (0.007)	0.103 (0.007)
$\mathrm{Variance}\;\sigma_{\varepsilon }{}^{2}$	0.179 (0.002)	0.177 (0.002)	0.176 (0.002)
Log likelihood	1836.415	1908.163	1935.25
AIC	−3640.829	−3760.326	−3798.5

Notes: *, ** and *** represent significance 10%, 5% and 1% level, respectively.

**Table 7 ijerph-19-05707-t007:** The estimated results of influencing factors of summer ***maize*** yield.

Variables	Model I	Model II	Model III
T_maize1_	−0.167 (0.111)	−0.158 (0.107)	−0.167 (0.104)
P_maize1_	−0.023 ** (0.011)	−0.013 (0.01)	−0.011 (0.010)
T_maize2_	0.533 *** (0.181)	0.427 *** (0.173)	0.453 *** (0.168)
P_maize2_	−0.016 * (0.009)	−0.012 (0.008)	−0.013 (0.008)
T_maize3_	−0.083 *** (0.03)	−0.047 (0.03)	−0.047 (0.029)
P_maize3_	−0.017 (0.012)	−0.013 (0.012)	−0.012 (0.011)
D_D_	−0.127 *** (0.041)	−0.13 *** (0.041)	−0.091 (0.080)
D_F_	−0.142 *** (0.043)	−0.138 *** (0.043)	−0.165 *** (0.043)
DL_D_	−0.136 *** (0.019)	−0.141 *** (0.019)	−0.489 *** (0.056)
DL_F_	−0.224 *** (0.027)	−0.219 *** (0.026)	−0.219 *** (0.026)
DL_R_	−0.122 *** (0.042)	−0.127 *** (0.042)	−0.133 *** (0.041)
DL_W_	−0.098 *** (0.023)	−0.101 *** (0.023)	−0.107 *** (0.023)
T_2011_	0.149 *** (0.036)	0.149 *** (0.035)	0.137 *** (0.035)
T_2012_	0.151 *** (0.021)	0.147 *** (0.021)	0.147 *** (0.021)
L_1_	−	0.108 (0.069)	0.094 (0.069)
L_2_	−	0.001 (0.047)	−0.009 (0.047)
L_32_	−	0.114 *** (0.024)	0.106 *** (0.024)
L_33_	−	0.147 *** (0.028)	0.145 *** (0.028)
ln(I_1_)	−	−0.006 (0.01)	−0.006 (0.010)
ln(I_2_)	−	0.032 *** (0.008)	0.033 *** (0.008)
ln(I_3_)	−	0.009 (0.007)	0.011 (0.007)
ln(I_4_)	−	−0.036 *** (0.013)	−0.036 *** (0.013)
ln(I_5_)	−	0.018 *** (0.003)	0.018 *** (0.003)
H_1_	−	0.0008 * (0.0004)	0.001 * (0.000)
H_2_	−	0.003 (0.003)	0.003 (0.003)
H_3_	−	0.033 (0.021)	−0.033 (0.020)
V_1_	−	−	−0.028 (0.023)
V_2_	−	−	−0.001 (0.001)
V_3_	−	−	0.004 (0.011)
V_1_ × D_D_	−	−	0.128 *** (0.022)
V_1_ × DL_D_	−	−	−0.11 *** (0.021)
V_2_ × D_D_	−	−	−0.001 (0.001)
V_2_ × DL_D_	−	−	0.005 *** (0.001)
V_3_ × D_D_	−	−	0.018 (0.012)
V_3_ × DL_D_	−	−	−0.018 (0.012)
Cons.	1.449 (2.052)	2.678 (1.977)	2.269 (1.93)
$\mathrm{Variance}\;\sigma_{\mu }{}^{2}$	0.152 *** (0.013)	0.141 *** (0.012)	0.133 *** (0.012)
$\mathrm{Variance}\;\sigma_{\varepsilon }{}^{2}$	0.541 *** (0.005)	0.537 *** (0.005)	0.532 *** (0.005)
Log likelihood	−4281.319	−4231.07	−4177.54
AIC	8596.638	8520.14	8431.08

Notes: *, ** and *** represent significance 10%, 5% and 1% level, respectively.

## Data Availability

Data supporting the conclusions of this article are included within the article. The dataset presented in this study are available on request from the corresponding author.

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
