# Peer review of "The Economic Impact of Climate Change on Wheat and Maize Yields in the North China Plain"

_ijerph, 2022, doi:10.3390/ijerph19095707_

Round 1

Reviewer 1 Report

The article has good potential for publication in that it highlights the interaction between several production factors, covering both economic and social perspectives. The statement of the problem was well laid out and argued. The researchers were able to illustrate winter wheat and maize production in the North China Plains were affected more by severe weather rather than long term climate change, thus giving importance to the short term management of crops, which was more under the control of farmers.

Nevertheless, the article lacks clarity in reporting the findings of the study. The sentence structure and choice of words made it difficult for the reader to understand the points the authors were trying to convey. For example, the authors indicated, “…the wheat yield decreased by 19.7% in the case of farmland plot drought, but it only decreased by 1% if the distance between the village committee and the nearest road above the township level reduced by 1 km.” (p. 10). It is possible this sentence that is heavily influenced most likely by local nomenclature, but most readers would have a hard time interpreting its meaning.

In the case of word choices, the authors for example used the term, ‘seedling emergency” at the beginning of the last paragraph on page 5. I believe the writers meant the emergence of shoots or seed germination. There are many more instances where phrases and words were rather confounding and can be improved through thorough editing. Hence, it is strongly recommended that the article be edited and rewritten for clarity.

Author Response

Dear Reviewer:

Thank you comments and suggestions. We have revised our manuscipt as you suggested on by one. 

Reviewer 2 Report

Well written paper.

Missing: limitations of the paper (methodology applied and data available) that would lead to further research.

Please check ithe editing rules of the Journal

Author Response

Dear Reviewer:

Thank your comments and suggestions. We have revised our manucript as you suggested and responded your commetns one by one. 

Reviewer 3 Report

The article presents an interesting study of the influence of climate change on the wheat and maize yields in the North China Plain. Despite employing several models and reaching relevant conclusions, the article has some serious shortcomings in terms of format and structure.

  1. The authors should redefine all citations and adapt them to the style of the journal.
  2. The authors should redefine all footnotes and adapt them to the style of the journal.
  3. Authors must redefine all tables, adapt them to the style of the journal and position them where they are cited.
  4. Authors should position figures where they are cited.
  5. Perhaps the most important fault of the article is that it completely lacks a section dedicated to discussion. We need to see what the article contributes in relation to other studies in its field.

Without these changes, the article should not be published.

Author Response

(The authors gave the same response as above.)

Round 2

Reviewer 3 Report

Thank you for considering all comments. I wish you the best of lucks with your paper.